# Cancer Angiogenesis and Opportunity of Influence on Tumor by Changing Vascularization

**DOI:** 10.3390/jpm12030327

**Published:** 2022-02-22

**Authors:** Igor Maiborodin, Alfija Mansurova, Alexander Chernyavskiy, Alexander Romanov, Vladimir Voitcitctkii, Anna Kedrova, Alexander Tarkhov, Alena Chernyshova, Sergey Krasil’nikov

**Affiliations:** 1The E. Meshalkin National Medical Research Center, Ministry of Health of Russia, 30055 Novosibirsk, Russia; a_mansurova@mail.ru (A.M.); amchern@mail.ru (A.C.); a-romanov@mehsalkin.ru (A.R.); v.vladimir2020@mail.ru (V.V.); kedrova.anna@gmail.com (A.K.); alexander-1976@rambler.ru (A.T.); professorkrasilnikov@rambler.ru (S.K.); 2Tomsk National Research Medical Center, Cancer Research Institute, Russian Academy of Sciences, 634009 Tomsk, Russia; alacher@list.ru

**Keywords:** angiogenesis, vasculogenesis, vascularization, vessels, cancer, tumor

## Abstract

Based on the study of recent scientific literature devoted to neovascularization and angiogenesis in malignant neoplasms, it was concluded that there are many publications on each of the problems of tumor angiogenesis and vascularization. The formation of blood vessels in a tumor and certain aspects of the prognostic value of the severity of vascularization in almost all forms of cancer are considered. Special attention is paid to the peculiarities of angiogenesis in tumors of the female reproductive system. A large number of vessels in the tumor often indicates a poor prognosis. The influence of various factors on the initiation of angiogenesis and the process itself, as well as the possibility of suppressing such signals to slow down the formation of blood vessels and thus the development of the tumor are widely studied. The results of pharmacological suppression of tumor vessel formation demonstrate a good clinical outcome but one accompanied by a large number of severe adverse side effects. Such a significant amount of studies on each of the problems of tumor vascularization indicates the increasing importance of this area of oncology. At the same time, only a very small number of works are devoted to the study of the differences in angiogenesis and number of vessels between different parts of the tumor, as well as between the primary tumor node and its metastases. The refinement of the results is still to be done. It was noted that the expression of proangiogenic factors in metastases is usually higher than in the source of metastasis, and the expression in lymphogenous metastases is higher than in hematogenous ones.

## 1. Introduction

In recent years, the treatment of ovarian cancer has been enriched with many new target treatment methods, particularly with antiangiogenics and PARP inhibitors (poly(ADP-ribose)polymerase—the enzymes that catalyze poly-ADP-ribosylation, one of the types of post-translational modification of proteins) that literally changed the natural course of the disease. Impressive results of immunotherapy at various gynecological neoplasms, such as cervical and endometrial cancer, have opened up opportunities for the introduction of immune-stimulating drugs and for the treatment of ovarian tumors [1].

Angiogenesis is an irreplaceable stage in the growth and invasiveness of a malignant tumor including a number of complex molecular stages. Increasing vascularization in cancer is necessary for its growth and metastatic spreading. Understanding the dominating factors that control tumor angiogenesis may reveal new therapeutic targets in cancer treatment [2,3,4]. Natural killer cells produce proangiogenic factors and participate separately or together with mast cells in the regulation of angiogenesis in both physiological and pathological conditions, including tumors [5].

Endotheliocytes originating from the endothelium of tumor blood vessels circulate along the bloodstream. It has been shown that such endothelial cells are associated with angiogenesis in the tumor and its growth and can determine the disease prognosis. Detection of CD44+ and vimentin+ endothelial cells in peripheral venous blood in patients with operable pancreatic adenocarcinoma prior to radical surgery may indicate a shorter period between surgery and tumor recurrence (<6 months) [6].

Although the link between the lymphatic system and the metastatic spread of cancer goes back centuries, the understanding of the underlying mechanisms of this interaction is still not definitively clarified. Lymphatic vessels provide a pathway for tumor cells to regional lymph nodes (LN) that is a prognostic of distant metastases and poor survival. Endothelial cells of lymphatic vessels respond to molecular signals from the microenvironment, mediating the growth and remodeling of lymphatic capillaries in the primary tumor site, draining LN and distant premetastatic niches. However, not only does this promote metastasis, it also affects antitumor immunity [7]. 

In connection with the above, the aim of this investigation was to provide a narrative overview of the literature regarding changes in cancer angiogenesis and vascularization to improve the results of treatment and to facilitate further research. The article briefly discusses vascularization as a predictor at cancer, the influence of VEGF on tumor growth and vascularization, results of tumor angiogenesis inhibition, tumor angiogenesis and stromal cells, and features of angiogenesis in ovarian cancer and other tumors of the female reproductive system.

## 2. Vascularization as a Predictor at Cancer 

Lymphangiogenesis in the lymph nodes has not been described, since in ontogenesis or during recovery from an injury, the lymph nodes originate from the lymphatic vessels, so angiogenesis in these organs is associated with an increase in intranodal blood vascularization only. It seems to us that the optimal marker for detecting newly formed blood vessels is the CD34 antigen. Antibodies to this antigen make it possible to detect only newly formed, “young” blood vessels. The state of vascularization in axillary LNs in patients with breast cancer (BC) was assessed by light microscopy using antibodies to the CD34 antigen. With the development of BC, regional LNs show a rapid and significant increase in the number of vessels. There is a strong or very strong positive correlation between the disease stage and the indicators of vascularization of the LN. At the same time, no significant differences in LN vascularization were found between women who have no cancer and women who have stage I tumors; the differences can be found only starting from stage II. What young, newly formed vessels in the nodes looks like is shown in Figure 1 [8,9].

Factors that contribute to tumor angiogenesis, for example, nestin, are usually combined with low specific and overall survival and a shorter period before recurrence [10]. Excessive production of interleukin-6 in patients is associated with the production of inflammatory cytokines and has a role in tumor angiogenesis that makes this release a potential aim for targeted therapy [11]. Angiogenesis and metastatic spreading are considered prognostic markers of BC; the transmembrane receptor tyrosine kinase is involved in both processes of tumor progression [12]. The expression of oxidoreductase which contains the WW domain was significantly lower in the tissue of cervical invasive squamous cell carcinoma compared with that in the benign tumor (leiomyoma) of this area (*p* = 0.019). The expression of such oxidoreductase was inversely proportional to the expression of CD31 in tumor samples (*p* = 0.018). This marker is associated with tumor angiogenesis [13].

A poor prognosis (low 5-year survival rate) with a high number of microvessels in the tumor interstitium was shown in 1121 patients with gastric cancer. Multivariate analysis showed that Lauren’s classification, depth of infiltration, nodal status, clinical stage, HER2 expression and number of microvessels are the independent factors influencing the prognosis in patients with gastric cancer (*p* < 0.05). In clinical practice, the targeted therapy against angiogenesis can be used for effective treatment [14].

AGTR-1 (angiotensin II receptor-1) which mediates signaling for vasoconstriction and inflammation in vascular disease is aberrantly overexpressed in some forms of BC. Overexpression of AGTR-1 correlated with aggressive features that include an increase in the frequency of metastasis in the LN, a decrease in sensitivity to neoadjuvant therapy, and a decrease in overall survival. Overexpression of AGTR-1 directs both ligand-independent and ligand-dependent activation of NF-κB (a universal transcription factor that controls the expression of immune response genes, apoptosis and cell cycle; dysregulation of NF-κB causes inflammation, autoimmune diseases, as well as the development of viral infections and cancer), mediated by a signaling pathway, the initiation of which causes internal reactions of cancer cells, including proliferation, migration, and invasion. In addition, NF-κB triggering affected the endothelial cells of the tumor microenvironment and promoted tumor angiogenesis. It is advisable to study the possibility of repurposing drugs that are oriented to act on angiotensin to improve the treatment of BC expressing AGTR-1 [15]. 

Expression of the NDRG3 protein promotes angiogenesis in the tumor and the growth of its cells. The expression of this protein was observed in 194 (14.5 %) cases of BC, and it was associated with age ≥ 50 years (*p* = 0.043), high histologic grade (*p* < 0.001), high Ki-67 (*p* < 0.001), low levels of estrogen and progesterone receptors (both *p* < 0.001), and positive HER2 status (*p* < 0.001). No significant correlation was found between NDRG3 expression and tumor size, LN status, lymphovascular invasion, or androgen receptor status. Association with lower overall survival was found in NDRG3 positive tumors (*p* = 0.035). Multivariate analysis showed that NDRG3 expression independently predicted overall survival (*p* = 0.011) and time to recurrences (*p* = 0.051) [16]. 

HOXB9 (homeobox protein Hox-B9) induces angiogenesis in a tumor, and it is associated with poor prognosis in patients with BC or colon cancers. The correlation between HOXB9 expression, prognosis, as well as clinical and pathological factors in patients with gastric cancer was assessed, and the contribution of HOXB9 expression to tumor cell lymphangiogenesis in vitro was analyzed. It was shown that the depth of tumor invasion, the number of metastases in the LN, and lymphatic and vascular invasion are strongly associated with the expression of HOXB9. Overall survival decreased in patients with high HOXB9 expression. Expression of VEGF-D mRNA, but not VEGF-C or VEGFR-3, was increased in tumor cells in vitro overexpressing HOXB9 compared with the checkpoint cells. In addition, the expression of HOXB9 positively correlated with the progression of gastric cancer and the expression of a lymphangiogenesis marker. HOXB9 may be associated with lymphogenous metastasis [17].

Exosomes derived from colorectal carcinoma cells with inhibited long noncoding RNA APC1 (anaphase-promoting complex subunit 1; the enzyme, which is one of at least 10 subunits of the anaphase stimulation complex which functions during the transition from metaphase to anaphase of the cell cycle and regulated by the spindle assembly checkpoint) promoted angiogenesis through MAPK activation (mitogen-activated protein kinase; can be activated by extracellular signals such as hormones, growth factors, chemokines and neurotransmitters that are recognized by the corresponding receptor tyrosine kinases; participates in T cell activation, proliferation of endothelial cells during increase in the number of blood vessels, in the regulation of synaptic plasticity, and phosphorylation of the transcription factor p53) in endotheliocytes [18].

Factors induced by hypoxia and a decrease of O_2_ availability activate the transcription of target genes encoding proteins that have an important role in communication between cancer and stromal cells. Various cancer cells were incubated under hypoxic conditions. All cells expressed mRNA and HIF-1α (hypoxia-inducible factor-1α) and HIF-2α proteins. However, the proliferation of non-small cell lung cancer, BC, stomach, and brain cancer under hypoxic conditions was more dependent on HIF-1α, except for hepatocellular carcinoma, where it was more dependent on HIF-2α. Among the HIF-1α-dependent cells, the H1299 line was most affected by the knockdown of HIF-1α in terms of cell proliferation. MDK (midkine—heparin-binding growth factor that promotes angiogenesis and carcinogenesis) significantly increased the migration of human umbilical vein endothelial cells and neovascularization of chorioallantoic membranes in chickens through paracrine signaling. Moreover, MDK secreted by NSCLC cells increased regulation of NF-κB and enhanced cancer formation. In response to MDK knockdown, siRNA (small interfering RNA), or MDK inhibitor, not only did MDK-induced endothelial cell migration and angiogenesis decrease, but the progression and metastasis of NSCLC cells in vitro and in vivo also discontinued. Treatment with MDK inhibitors significantly increased survival of mice compared with the control group or the MDK expression group [19]. 

The correlation between the HGF expression (hepatocyte growth factor) and FAP (fibroblast activation protein) with an increase in the number of blood vessels and metastasis in colorectal cancer (127 samples), colorectal polyps (51 cases), and unaltered tissues (28 cases) was studied. The expression of HGF and FAP was significantly higher in the group with oncological disease than in the groups with sound tissues and polyps (*p* < 0.05). Moreover, positive indices of HGF and FAP expression in the group with LN metastases were clearly higher than in the group with nonlymphatic metastasis (*p* < 0.05). The expression of HGF and FAP was higher in the group with liver metastases than in the group with nonhepatic metastases. The group with colorectal cancer had a much higher microvessel density compared to the groups without tumors (*p* < 0.05). At positive HGF and FAP values, the number of vessels exceeded this indicator. At a negative level of these factors (*p* < 0.05), the content of HGF and FAP positively correlated with vascularization (r = 0.542 and *p* < 0.001; r = 0.753 and *p* < 0.001, respectively). FAP and HGF have an important role in increasing vascularization in patients with colorectal cancer, and their expression levels are significant for predicting liver and LN metastases [20].

The expression of HSF-1 (heat shock factor-1) in patients with pancreatic cancer was studied, and the relationship between HSF-1, an increase in the number of blood vessels, clinical pathological factors, and prognosis was determined. Compared with sound pancreatic tissue and peritumoral tissue, HSF-1 RNA and the protein itself showed considerably higher expression in pancreatic cancer tissue and were strongly associated with microvascular density. High expression of HSF-1 did not correlate with the sex and age of patients, the level of carcinoembryonic antigen, or the diameter and location of the tumor. However, it was statistically significant consistent with LN metastases, stages of metastases, degree of differentiation, vascular invasion, and distant metastases. The expression level of HSF-1 was positively correlated with prognosis and survival. High HSF-1 expression indicates poor prognosis, reduction in life expectancy, and decrease of delaying time to recurrences [21].

Paxillin (PXN), a key component of the focal adhesion complex, is associated with cancer progression. PXN present in the cell nucleus enhances angiogenesis through transcriptional regulation of SRC expression (not associated with the cell receptor a tyrosine kinase involved in the processes of embryonic development and cell growth), which, in turn, increases the expression of PLAT (tissue plasminogen activator) due to the activation of NF-κB in endothelial cells that promotes angiogenesis. PLAT is a protein belonging to the group of secreted proteases that converts the proenzyme plasminogen into its active form, plasmin, which is a fibrinolytic enzyme. Hyperactivation of the enzyme leads to excessive bleeding, and the decreased activity leads to inhibition of fibrinolysis processes that can lead to thrombosis and embolism. Suppression of PXN in ovarian cancer mouse models reduces the number of blood vessels, tumor growth, and metastatic spreading [22]. 

The general blood flow in the tumor and its heterogeneity in newly diagnosed BC (217 patients) according to the biological characteristics of the tumor and the molecular subtypes was studied. The interaction between perfusion and metabolism was also analyzed. A higher level of perfusion was shown in tumors with LN lesions. There were no significant differences in total blood flow values depending on tumor metabolism. The heterogeneity of perfusion correlated well with the heterogeneity of metabolism; therefore, it is possible that blood flow and consequently, tumor angiogenesis can cause heterogeneity of metabolic processes [23].

## 3. Influence of VEGF on Tumor Growth and Vascularization

VEGF may have an important role in tumor progression and metastatic spreading as well as in cancer angiogenesis [4,24,25,26,27,28,29,30]. One of the regulators of VEGF-A is the NFE2L3 nuclear transcription factor which is associated with a poor prognosis in pancreatic cancer. A decrease in its level reduces metastatic spreading in the LN [31].

The number of blood vessels in cervical tumors in 67 patients was assessed by immunohistochemical staining using antibodies to CD31 and VEGF. High tumor vascularization is an unfavorable prognostic parameter for cervical cancer, indicating low overall survival (*p* = 0.004) and disease-free survival [32].

Inhibition of tumor angiogenesis may be a new target in the treatment of colorectal cancer. When examining 63 paraffin blocks with primary tumor samples, the level of VEGF expression was 84.1%. VEGF was markedly associated with histopathological type (*p* = 0.01) and invasion (*p* = 0.02) of the tumor and with the degree, metastases in the LN, and stage (*p* < 0.001) of the disease. A positive relationship was found between VEGF expression and fatty acid synthase in colorectal cancer (*p* < 0.001). It is quite likely that fatty acid synthase is a positive antiangiogenic target [26].

Oral squamous cell carcinoma is an aggressive tumor with a poor prognosis and a high level of local invasion and metastases in the LN. VEGF-A has an important role in the angiogenesis and metastasis of this tumor. The high level of VEGF-A expression positively correlated with the stage of the disease in patients with carcinoma. MCP-1 (CCL2; monocyte chemotaxis factor) in mammals controls the egress of cells from hematopoietic organs and their traffic to inflammatory focus. In cancer cells MCP-1 increases the expression of VEGF-A and promotes angiogenesis. MicroRNA-29c (miR-29c) mimics the reverse activity of MCP-1. MCP-1 is a new molecular therapeutic target for the inhibition of angiogenesis and metastasis of oral squamous cell carcinoma [33].

Biopsy samples from 117 patients with esophageal squamous cell carcinoma were studied using immunohistochemistry. None of the patients had distant metastases before surgery, and they did not undergo preoperative chemotherapy or radiotherapy. Patients with a high expression of VEGFR-2 and a high level of tumor vascularization were characterized by a lower five-year survival rate (*p* < 0.05) [24]. About 70% of the 408 examined BC patients had high VEGF-C expression which was strongly associated with severe tumor stages (*p* = 0.019), significant Ki67 proliferation index, LN metastases (N3) and lymphatic vascular invasion in univariate analysis [4].

The expression of VEGF in the benign tumor (pleomorphic adenoma) and two malignant tumors (mucoepidermoid carcinoma or adenoid cystic carcinoma) of salivary glands of various structures was analyzed. There was a statistically significant difference in the expression and activity of VEGF between these tumors (*p* < 0.05). However, there was no significant correlation between the parameters of this factor and the histological grade and metastases in the LN at malignant processes [27].

AEG-1 (astrocyte elevated gene-1) mediates angiogenesis, and enhancement of the functions of this gene is responsible for tumor angiogenesis during cancer formation. AEG-1 expression, VEGF, and the density of intratumoral microvessels was studied in 88 paired tumor samples and adjacent sound tissue samples obtained from patients with non-small cell lung cancer. Overexpression of AEG-1 was observed in 61.3% of tumor samples compared with 6.8% (6/88) of sound tissues (*p* < 0.001). Expression of AEG-1 in such cancer was associated with a high stage of TNM (*p* = 0.021), tumor de-differentiation (*p* = 0.034), vascular invasion (*p* = 0.035), metastases in the LN (*p* < 0.001) and poor overall survival (*p* = 0.024). Moreover, AEG-1 expression was associated with tumor angiogenesis with VEGF overexpression (*p* < 0.001) and intratumor microvascular density (*p* < 0.001). AEG-1 probably has an important role at the level of transcription in malignant transformation and increasing vascularization in tumors in this form of cancer, and the expression of anti-AEG-1 mRNA may be a potential strategy for antiangiogenetic therapies. The adjuvant therapy with an antiangiogenic agent should be used in the early postoperative period prior to the initiation of conventional chemotherapy in patients with non-small cell lung cancer and AEG-1 overexpression [34,35].

Integrins are a large family of adhesion molecules that mediate cell-to-cell cooperation and interaction. It was found that many of 24 integrin isoforms are associated with tumor angiogenesis, migration and proliferation of cancer cells, and metastasis. Integrins, especially αvβ3, αvβ5, and α5β1, are involved in mediating vascular development in tumors by interacting with VEGF and Ang-TIE (angiopoietin tyrosine kinase: membrane protein, Ang receptor) signaling pathways [36]. 

How tumor cells induce angiogenesis is shown schematically in Figure 2.

## 4. Results of Tumor Angiogenesis Inhibition

### 4.1. Effects of VEGF Suppression and Inactivation of Its Receptors 

Proangiogenic factors, including VEGF, have been recognized as key therapeutic targets in cancer treatment [4,24,25,27,28,29,30]. 

The use of antiangiogenic agents (apatinib) for the treatment of malignant processes, targeting the VEGF-2 receptor and thereby suppressing tumor angiogenesis, is effective for the therapy of patients with sarcoma, cancer metastases in the brain, and peritumor brain edema [28,29]. Highly vascularized tumors in patients with clear-cell renal carcinoma express high levels of VEGF. The positive result of using a drug with neutralized antibodies to this cytokine (bevacizumab) has been described [30].

The level of miR-129-5p in lung cancer cells is much lower than in sound tissues, and the difference is statistically significant. Compared to the patients with highly expressed miR-129-5p, the patients with low levels had higher rates of LN involvement or distant metastases, and overall survival was lower. VEGF expression negatively correlates with miR-129-5p. After the use of miR-129-5p mimetics, the cell proliferation and invasiveness and migration as well as the activity of tumor angiogenesis sharply decreased in comparison with cells with no impact. VEGF overexpression in the experiment counteracted the effect of miR-129-5p mimetics on angiogenesis in the tumor as well as on the invasive and migratory capacity of lung cancer cells that together lead to the progression of the malignant process. The effect of miR-129-5p is most likely related to the ability to regulate VEGF [37].

The effects and mechanisms of oxyresveratrol (oxyres) action on the development of hepatocellular carcinoma in vitro and in vivo were investigated. The drug significantly suppressed the proliferation and migration of QGY-7701 and SMMC-7721 cells and inhibited the growth of the transplanted tumor (*p* < 0.001) as well as the metastasis to sentinel LNs (70%) in a dose-dependent manner in the experiment. In addition, the density of blood and lymph microvessels in the tumor was significantly reduced (*p* < 0.05), and the expression of CD31, VEGFR-3, and VEGF-C was inhibited (*p* < 0.05). Oxyres has an antitumor effect in hepatocellular carcinoma by suppressing both angiogenesis and metastasis in the LN [38].

PHD-3 (prolyl hydroxylase-3) is widely used as a tumor suppressor. Expression of PHD3 was increased in sound tissue adjacent to the tumor as compared to gastric cancer tissue, and overexpression of PHD-3 correlated with the presence of well-differentiated cancer cells, early stage of cancer, and the absence of metastases in the LN. The experiments in vitro demonstrated that PHD-3 can act as a downregulator of HIF-1α and VEGF that are involved in tumor angiogenesis [39]. 

### 4.2. Suppression of Factors Indirectly Associated with Angiogenesis

The Ang/TIE-2 interaction has a key role in tumor angiogenesis. The increasing vascularization in a tumor is modulated at both the epigenetic and protein levels and has potential implications for the response of immune cells. In human cholangiocarcinoma samples, the expression of angiogenesis-associated miRs, Angs, and monocytes expressing TIE-2 as an Ang receptor, was assessed in terms of prognostic value after hepatectomy. MiR-126 was suppressed in 76.7% of all tumor samples. High relative expression was associated with smaller tumors and the decreased metastases in the LN. High Ang-1 expression was associated with less carcinomatous lymphangiosis and a better histological score (total *p* < 0.05). High miR-126, low miR-128, and TIE-2 expressing monocytes were independent predictors of relapse free survival and overall survival (total *p* < 0.05). The absence of such monocytes correlated with an increased level of the tumor CA19-9 marker used in the diagnosis of pancreatic cancer [3,40,41].

B7-H3 (CD276; has a role in the regulation of the T-lymphocytic immune response, a role in increasing the survival of tumor cells through inhibition of cell lysis initiated by natural killer cells) is a new member of the B7 costimulatory molecule family. It performs a critical function in the T-cell mediated anticancer immune response. Abnormal expression of tumor B7-H3 is often associated with poor prognosis, and B7-H3 can serve as an effective endothelial marker for the prognosis of the course of some types of cancer. It is also a potential target for antivascular therapy [3,41]. 

B7-H3 is not expressed in liver tissues, but it is found in 57.8% (26/45) of human cholangiocarcinoma cases. Expression of B7-H3 is strongly associated with LN metastases and venous invasion. Positive correlation was observed between the expression of B7-H3 and the number of microvessels, an index of tumor angiogenesis. Patients with negative B7-H3 expression had higher overall survival and specific survival rates than the patients with positive expression [41].

B7-H3 is highly expressed in the vascular endothelium of renal clear-cell carcinoma, and it is associated with the degree and stage of tumor node metastasis (TNM). Microvessel number density also correlated with tumor grade and TNM stage. B7-H3 and TIE-2 expressions positively correlated with vascularization of the tumor node. In addition, immunofluorescent staining revealed coexpression of B7-H3 and TIE-2 in the vascular endothelium of clear-cell renal carcinoma. It is possible that the expression of B7-H3 and TIE-2 in the vasculature of renal carcinoma is closely related to the progression and prognosis of the disease. It is also possible that B7-H3 promotes angiogenesis via the TIE-2 pathway [3].

Suppression of SOCS-6 (an inhibitor of cytokine-6 signaling) correlated with malignant progression of human prostate cancer, and it was associated with late clinical stage (*p* = 0.029) and metastasis in the LN (*p* = 0.013). SOCS6 overexpression inhibited the invasion, the migration of prostate cancer cells, the growth of tumor xenotransplants, and angiogenesis. Induction of apoptosis (*p* < 0.05) coincided with suppression of Bcl2 and Hspa1a, and suppression of tumor angiogenesis coincided with suppression of F7, Fak3 and Frzb (involved in the regulation of growth and differentiation of certain cell types) [42].

Ang-2 suppression and improvement of pericyte covering of melanoma vessels with the use of diosmetin (a natural flavonoid obtained from citrus fruits and some medicinal herbs) in an experiment with mice resulted in suppression of the formation of metastases in the lungs and LN. The drug caused the death of tumor cells (induction of apoptosis of B16F10 melanoma cells through the caspase pathway) and the pronounced inhibition of tumor angiogenesis (suppression of vascular invasion and engorgement of tumor vessels in vivo, proliferation of rat aortic rings by endotheliocytes ex vivo, formation of tubes and migration of human endothelial cells in vitro), and normalized the defective tumor vasculature [43].

Baicalein, a herbal remedy, is a natural flavonoid derived from the *Scutellaria baicalensis* Georgi roots. The drug is known for its anticancer, anti-inflammatory and neuroprotective properties. B16F10 cells, Lewis lung carcinoma cells, and human umbilical vein endotheliocytes were used to study the effect of *baicalensis* on cell proliferation and viability, migration, and tube formation in vitro. In addition, an animal model was used to monitor the growth rate and metastasis of tumors and the formation of tumor vessels in vivo. It was found that *baicalensis* reduces proliferation and migration and induces the death of tumor cells due to the activation of caspase-3 in B16F10 cells and Lewis carcinoma and strongly inhibits the formation of tubes and migration of endotheliocytes. *Baicalenisis* reduced tumor volume in an experiment with mice and decreased the cancer growth rate in the early stages. The groups with the *baicalensis*-treated patients showed significantly reduced expression of endothelial cell markers CD31 and α-SMA (smooth muscle actin protein found in almost all mammals) in tumors indicating that the drug inhibits tumor angiogenesis, disrupting the development of the vasculature. Evaluation of LN and lung samples showed that *baicalensis* reduces metastasis to those tissues [44]. 

## 5. Tumor Angiogenesis and Stromal Cells 

Tumor progression begins when cancer cells recruit tumor-associated stromal cells to form vessels, and this ultimately leads to uncontrolled growth, invasion, and metastasis. Induced mesenchymal stem cells (MSCs) with a pronounced angiogenic potential and expressing high levels of HIF-1α and proangiogenic factors belonging to the VEGF, PDGF (platelet-derived growth factor) and Ang subfamilies were created. Coinjection of these MSCs and 4T1 BC cells into the fatty pads of the mouse mammary gland caused highly aggressive tumor growth (a 2-fold increase in tumor volume compared to some cancer cells, *p* = 0.01283) and a higher spontaneous metastatic spread in the lungs [45].

Tumor pericytes and other perivascular cells are commonly present in the stromal compartment of various human solid tumors and cancer xenografts in rodents in experiments. Tumor pericytes migrate to the LN in human oral cancer. In rat brain glioblastoma U-251 xenografts, pericytes originating from a human tumor covered rat-derived endothelial cells to form a mosaic of newly formed blood vessels and differentiate directly into tumor cells. The effect was expressed in the continuation of the production of glioblastoma tumor cells from malignant pericytes. In total, it can be assumed that tumor pericytes can give rise to tumor cells and serve as a potential source of cells for distant metastases [2]. 

Tumor formation from injected cells and grafts of bladder urothelial cell carcinoma in NOD/SCID mice, angiogenesis and metastasis to distant organs was significantly enhanced in the presence of LN stromal cells [46].

## 6. Features of Angiogenesis in Ovarian Cancer and Other Tumors of the Female Reproductive System

Ovarian cancer is a malignant tumor that quickly leads to death if left untreated. Therapies and treatments for ovarian cancer are constantly expanding while achieving sustainable clinic results remains largely of concern [47]. 

The features of blood and lymphatic vascularization and their significance for prognosis in ovarian cancer were investigated. The study included 139 women with epithelial ovarian tumors: 86 malignant, 17 borderline, and 36 benign. The density, percentage, average size, and number of blood vessels were determined using antibodies against CD34 and CD105 antigens in tumors. The parameters of lymphatic vessels were obtained using antibodies D2-40 to podoplanin. The increasing vascularization was most abundant in malignant tumors. Small lymphatic vessels predicted a 26% reduction in 5-year survival. In addition, a high percentage of lymphatic vascularization in tumors was associated with metastases in the LN, and the high density was associated with cancer recurrence. Lower numbers of microvessels, as assessed by CD34 staining, predicted shorter progression-free survival. Additionally, the large size of microvessels measured by CD34 and the high number of vessels assessed by CD105 were related to residual tumor >1 cm at primary surgery and large vessel size, as assessed by CD105 staining, was associated with stage III. Thus, CD34 and CD105 antigens determine different characteristics of blood vascularization. Lymphatic vascular parameters may predict prognosis in patients with ovarian cancer [48].

The Notch signaling pathway is involved in new vessel formation through the regulation of process growth [49]. The Notch signaling pathway is present in most animals. The Notch signaling pathway is important for intercellular cooperation, which includes regulatory mechanisms for genes that control multiple cell differentiation processes during fetal and adult life. Notch signaling is not regulated in many cancerous processes, and abnormal transmission is implicated in many diseases, including T cell acute lymphoblastic leukemia [50], multiple sclerosis, and many other disease states. Inhibition of Notch signaling has been shown to have an antiproliferative effect on T cell acute lymphoblastic leukemia in cultured cells and in a mouse model [51,52]. It was also found that an inhibitory effect on Notch expression in MSCs prevents differentiation [53]. This process is important for both normal ovarian angiogenesis and tumor vascularization, and it is regulated by Notch-VEGF cross-links. Moreover, Notch was associated in the ovaries with stem cell maintenance and epithelial-mesenchymal transition. Dysregulation of the Notch pathway is common in ovarian cancer, and it is associated with the impaired survival, advanced stages, and LN involvement. Notch also has a role in chemical resistance to platinum drugs [49].

Antiangiogenic therapy, a standard treatment for ovarian cancer, has variable efficiency. In addition, little is known about the prognostic biomarkers and factors affecting angiogenesis in cancer tissue. In this regard, Ang-2 expression and two endothelial TIE-1 and TIE-2 tyrosine kinase receptors were determined and their significance for predicting survival of patients with ovarian cancer was evaluated. The study also compared the expression of these factors between primary serous tumors of high grade and their distant metastases. A group of 86 women with primary epithelial ovarian cancer were examined. Distal omental metastases were found in 18.6% of cases (*N* = 16). The expression level of angiogenic factors was assessed by immunohistochemistry (306 samples) and real-time polymerase chain reaction (111 samples). A high level of epithelial TIE-2 expression is an important prognostic factor in high-grade primary serous ovarian cancer. Such expression predicts significantly shorter overall survival in both univariate (*p* < 0.001) and multivariate analyses (*p* = 0.022). Low levels of Ang-2 expression in primary ovarian tumors were strongly associated with shorter overall survival (*p* = 0.015) in univariate analysis, and they were also strongly associated with high-grade tumors, residual tumor size after primary surgery, and cancer recurrence (*p* = 0.008; *p* = 0.012 and *p* = 0.018, respectively) in the entire study sample. Ang-2 and TIE-2 expression was stronger in distal omental metastases than in high-grade primary serous tumors when analyzed of paired values (*p* = 0.001 and *p* = 0.002, respectively). Ang-2 expression levels and its TIE-2 receptor increase in metastatic lesions compared with primary tumors [54].

Most women with advanced ovarian cancer respond well to a treatment consisting of surgical resection and ≈ 6 cycles of platinum-based chemotherapy. Nevertheless, many patients have a disease recurrence, and there is a need for after-treatment. Targeted therapy with the angiogenesis inhibitor (bevacizumab) and the PARP inhibitors (olaparib, niraparib, and rucaparib) have shown significant clinical benefits as maintenance treatment for recurrent disease. Side effects associated with bevacizumab include hypertension, proteinuria, and noncentral nervous system hemorrhages, while the use of PARP inhibitors is associated with nausea, vomiting, fatigue, and anemia. Complications and adverse side effects of angiogenesis inhibitors have a significant impact on the patient’s health, even considering the toxicity associated with chemotherapy [55,56,57]. 

To assess the efficiency and toxicity of angiogenesis inhibitors for the treatment of ovarian cancer patients, C. Guo et al. [58] made a meta-analysis with data from 11,254 patients published before 12 August 2019. The data showed that therapy with angiogenesis inhibitors can significantly improve progression-free survival and overall survival in patients with ovarian cancer. However, this therapy is associated with a higher risk of common grade ≥3 adverse events such as arterial thromboembolism, ascites, diarrhea, gastrointestinal perforation, headache, hypertension with hemorrhages, hypokalemia, leucopenia, pain, proteinuria, thrombocytopenia, and thrombosis or embolism. That is, angiogenesis inhibitors can significantly improve progression-free survival and overall survival in patients with ovarian cancer, and at the same time they can increase the incidence of common adverse events.

The safety and efficiency of the combination of oral tyrosine kinase inhibitor (nintedanib) with oral cyclophosphamide in patients with recurrent ovarian cancer were investigated. The study involved 117 women with an average age of 64 with recurrent ovarian cancer, fallopian tube cancer, or primary peritoneal cancer. Grade 3/4 adverse events were observed in 64% (nintedanib) versus 54% (placebo) of patients (*p* = 0.28): lymphopenia (18.6% nintedanib versus 16.4% placebo), diarrhea (13.6% versus 0%), neutropenia (11.9% versus 0%), fatigue (10.2% versus 9.1%) and vomiting (10.2% versus 7.3%) were the most common toxicities. The time on treatment was reduced by 52 days (*p* < 0.01) in patients previously treated with bevacizumab. Of the sample, 26 patients (23%) took oral cyclophosphamide during ≥ 6 months. No differences in quality of life were found between treatment groups. Nintedanib did not improve results when it was added to oral cyclophosphamide [59]. 

It was studied whether blocking several points of the angiogenesis pathway by adding sorafenib (a multikinase inhibitor of VEGFR-2/3 and PDGFR) to bevacizumab, would show clinical activity in ovarian cancer. The study involved 54 women, 41 of whom had not previously taken bevacizumab; 13 had already received it. Treatment-related side effects of grade 3–4 (≥5%) included arterial hypertension and venous thrombosis or pulmonary embolism. The result of the combination of bevacizumab and sorafenib did not meet the predetermined target, but some clinical activity was observed in patients with platinum-resistant disease who were first intensively treated with bevacizumab. Toxicity correction requires careful monitoring and dose modification [60].

New therapeutic strategies for ovarian cancer are urgently needed due to the development of resistance and the adverse side effects of platinum-based chemotherapy. Theasaponin E1 is an oleanan-type saponin from the *Camellia sinensis* seeds. Activity of theasaponin concerning induction of apoptosis, cell cycle arrest, and suppression of angiogenesis in platinum-resistant ovarian cancer cells was determined in vitro. The results showed that this saponin has a stronger inhibitory effect on the growth of ovarian cancer OVCAR-3 and A2780/CP70 cells than cisplatin, and it is less cytotoxic to normal ovarian cells IOSE-364. Theasaponin strongly induced apoptosis of OVCAR-3 cells through internal and external pathways, slightly delayed the mitotic cycle in the G2/M phase, and inhibited OVCAR-3 cell migration and angiogenesis through a decrease in VEGF secretion and expression. Theasaponin E1 can be a prospective candidate to improve treatment of platinum-resistant ovarian cancer [61].

The mechanism by which MSC exosomes influence the development of ovarian cancer was researched. MYB (proto-oncogene, a transcription factor) was highly expressed in ovarian cancer, and exosomal miR-424 was poorly expressed. It is important that MYB has been identified as a target gene for miR-424. In addition, it was found that the transfer of miR-424 by MSC exosomes suppresses proliferation, migration, and invasion of ovarian cancer cells with a decrease in VEGF and VEGFR expression. MSC exosomes with miR-424 overexpressing can inhibit proliferation, migration, and tube formation by human umbilical vein endotheliocytes, as well as suppress tumor genesis and angiogenesis in the ovaries in vivo [62].

NGF (nerve growth factor) increases its expression during the progression of epithelial ovarian cancer, promoting cell proliferation and increasing vascularization through several oncogenic proteins such as c-MYC and VEGF. The expression of these proteins is controlled by miR-145, the dysregulation of which is associated with the tumor. MiR-145 overexpression reduces proliferation, migration, and invasion of epithelial ovarian cancer cells that was associated with a decrease in the levels of c-MYC and VEGF proteins. A decrease in tumor formation and inhibition of metastasis was observed in mice injected with ovarian cancer cells using miR-145 overexpression. Ovarian cell lines stimulated by NGF decreased the transcription and content of miR-145-5p. The increase of miR-145 regulation can be used as a therapeutic strategy in the treatment of epithelial ovarian cancer [63].

MiR-365 overexpression leads to a decrease in the proliferation rate and formation of clones of human ovarian cancer cells, as well as inhibition of angiogenesis and suppression of VEGF, Ang-1 and MMP-2 expression [64]. 

Nectin-2 is an adhesion molecule that has a role in tumor growth, metastasis, and angiogenesis. The level of nectin-2 was significantly increased in tumor biopsies from patients with ovarian cancer with metastases in the LN and with residual tumor > 1 cm after surgery. Nectin-2 expression was strongly suppressed in the peritoneal endothelium that was associated with a significant increase in serum VEGF levels. In cell culture, exposure to VEGF resulted in a significant decrease in nectin-2 levels that could be reversed by inhibition of VEGF. Knockdown of nectin-2 in endothelial cells was associated with a significant increase in endothelial permeability. Expression of nectin-2 in ovarian cancer can maintain adhesion of tumor cells that leads to growth and metastasis in the LN, and VEGF-induced suppression of nectin-2 in the peritoneal endothelium can increase vascular permeability leading to the formation of ascites [65].

Estrogens and androgens have an important role in sound and cancer tissue and down-regulate the expression of PEDF (pigment epithelium-derived factor, known as serpin F1, a multifunctional secreted protein with anti-angiogenic, antitumor, and neurotrophic functions) in tumors that are sensitive to sex hormones. PEDF inhibits tumor growth, and PEDF expression by estrogen influences oncogenesis, metastasis, and progression. PEDF expression is reduced in prostate, BC, ovarian, and endometrial tumor tissues compared to their counterparts from sound tissues, while an association between PEDF suppression and the effects of sex hormones in the time of preclinical studies was observed. PEDF reduces the growth and metastasis of tumor cells promoting apoptosis, inhibiting angiogenesis, increasing adhesion, and decreasing migration. PEDF can also prevent treatment resistance in some forms of cancer by inhibiting signaling to estrogen receptors. By interacting with components of the tumor microenvironment, PEDF counteracts the proliferative and immunosuppressive effects of estrogens, ultimately reducing tumorigenesis and metastasis. [66]. 

Aromatase inhibitors such as anastrozole, letrozole, and exemestane prevent metastasis and angiogenesis in BC and ovarian tumors that are positive for estrogen receptors. Inhibitors react primarily by reducing the production of estrogen in postmenopausal patients. Unfortunately, modern therapies based on aromatase inhibitors often have harmful side effects: acquired resistance, increased cancer recurrence rates, etc. There is an urgent need to identify new drugs in this group with fewer side effects and improved therapeutic efficiency [67]. 

Proteases have a critical role in the progression and metastasis of ovarian cancer. The degradation and fragmentation of the pericellular protein, along with the remodeling of the extracellular matrix, is carried out by numerous proteases that are present in the microenvironment of the ovarian tumor. In addition, the increased activity of certain proteases, including matrix metalloproteinases, affects tumor angiogenesis and the development of metastatic implants [68].

## 7. Conclusions

As noted throughout, the scientific literature of recent years includes many publications on each of the problems of tumor angiogenesis. The features of the formation of blood vessels in a tumor and certain aspects of the prognostic value of the severity of vascularization in almost all forms of cancer have been considered. Special attention has been paid to the peculiarities of angiogenesis in tumors of the female reproductive system. A large number of vessels in the tumor often indicates a poor prognosis. The influence of various factors on the initiation of angiogenesis and the process itself, as well as the possibility of suppressing such signals to slow the formation of blood vessels and thus, the development of the tumor, are widely studied. The results of pharmacological suppression of tumor vessel formation demonstrate a good clinical outcome, but, unfortunately, are accompanied by a large number of severe adverse side effects. Such a significant amount of studies on each of the problems of tumor vascularization indicates the increasing importance of this area of oncology. At the same time, only a very small number of works are devoted to the study of the differences in angiogenesis and number of vessels between different parts of the tumor, as well as between the primary tumor node and its metastases. The refinement of the results is still to be undertaken. It was noted that the expression of proangiogenic factors in metastases is usually higher than in the source of metastasis, and the expression in lymphogenous metastases is higher than in hematogenous ones. 

## Figures and Tables

**Figure 1 jpm-12-00327-f001:**
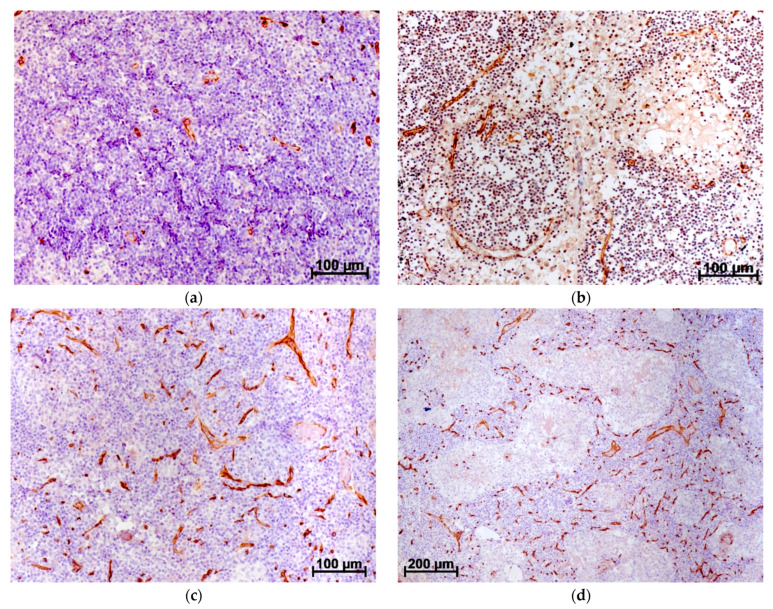
Human axillary LN. Immunohistochemical reaction with CD34 antibodies stained with diamidobenzidine and hematoxylin: (**a**) control LN, few CD34+ vessels in the superficial part of the cortex and paracortex, waste areas of the cortical substance free from CD34+ structures; (**b**) control LN, due to the postmortem changes, the medullary cords and sinuses lost well-defined structure, cell composition rare, single CD34+ vessels located at the edge of cords and sinuses, no CD34+ structures observed in the huge areas of the medullary substance; (**c**) LN at BC stage IIIa, small CD34+ vessels infiltrate practically the total cortical substance, walls of these vessels consist of one cell layer. (**d**) LN at BC stage IIIa, small CD34+ vessels infiltrate practically the total medullar substance of parenchyma [8,9].

**Figure 2 jpm-12-00327-f002:**
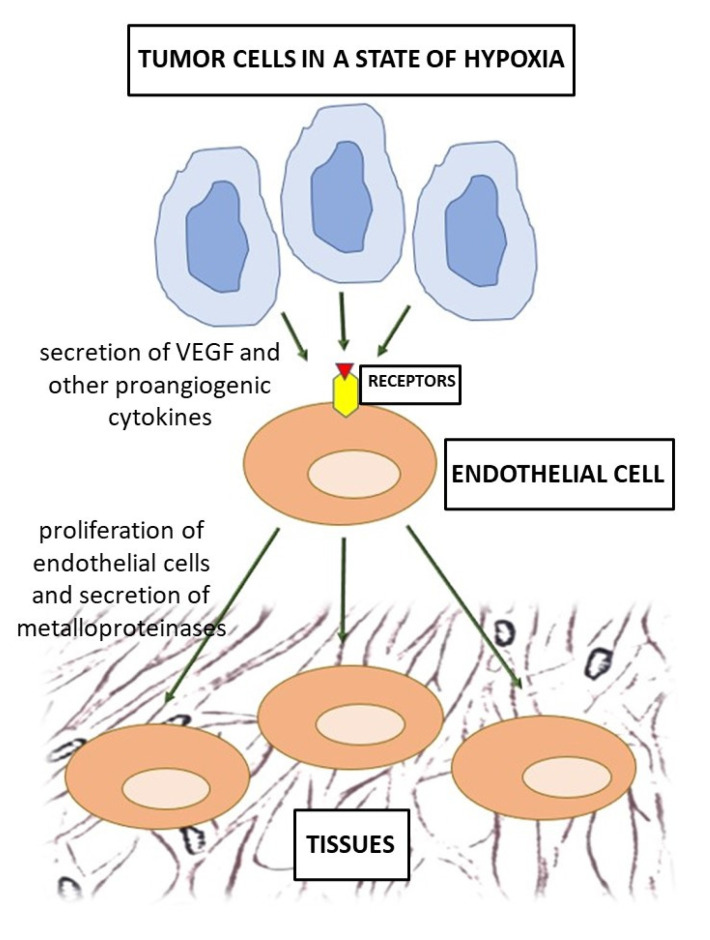
Schematic of the tumor angiogenesis.

## Data Availability

The data presented in this study are available on request from the corresponding author.

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
