# Peer review of "Cancer Angiogenesis and Opportunity of Influence on Tumor by Changing Vascularization"

_jpm, 2022, doi:10.3390/jpm12030327_

Round 1

Reviewer 1 Report

The reviewed paper summarizes the knowledge about the  tumour angiogenesis and influence on vascularization. There is quite large amount of literature on the topic, but the comprehensive review of present knowledge is necesasary. The paper is  interesting. However, in my opinion, it needs some minor corrections before publication. The list of my notes are below.

  1. There is lack of clear written aim of the paper. The Authors should strongly underline the importance of the described topic
  2.  The sentence "Various cancer cells.." shuld be revised without all these cancer cell lines.
  3.  The word: „NFκB” should be written NF-κB
  4.  The word: „Natural-killer cells” should be written Natural killer cells.
  5. Conclusion should be revised by the Authors very carrefully and written in straightforward. 

To conclude, in my opinion, the paper may be published in „TUMOR ANGIOGENESIS AND POSSIBILITY OF INFLUENCE ON TUMOR BY VASCULARIZATION CHANGING” after minor revision.

Author Response

I want to thank the Dear Reviewer for the short and clear review. On the substance of the remarks made:

  1. There is lack of clear written aim of the paper. The Authors should strongly underline the importance of the described topic

Corrected. Research aim added.

  1. The sentence "Various cancer cells.." shuld be revised without all these cancer cell lines.

Corrected.

  1. The word: „NFκB” should be written NF-κB

Corrected. That's what it writes in the original article: Ekambaram P, Lee JL, Hubel NE, Hu D, Yerneni S, Campbell PG, Pollock N, Klei LR, Concel VJ, Delekta PC, Chinnaiyan AM, Tomlins SA, Rhodes DR, Priedigkeit N, Lee AV, Oesterreich S, McAllister-Lucas LM, Lucas PC. The CARMA3-Bcl10-MALT1 Signalosome Drives NFκB Activation and Promotes Aggressiveness in Angiotensin II Receptor-Positive Breast Cancer. Cancer Res. 2018 Mar 1;78(5):1225-1240. doi: 10.1158/0008-5472.CAN-17-1089. Epub 2017 Dec 19. PMID: 29259013; PMCID: PMC6436094.

  1. The word: „Natural-killer cells” should be written Natural killer cells.

Corrected.

  1. Conclusion should be revised by the Authors very carrefully and written in straightforward. 

I'm never 100% sure of the correct interpretation of the results. Therefore, I try to write "possibility", "maybe", etc. We tried to fix it.

I hope that we have answered all the questions and comments of the Dear Reviewer.

Reviewer 2 Report

The title and parts of the manuscript suggest a review on tumorangiogenesis and especially its spacial components within a tumor and a comparision between primary tumor and metastasis. The manuscript includes sections on many factors involved in tumorangiogenesis.

The abstract makes this revier expect a focus on the spacial aspects of angiogenesis within a tumor and between tumor and metastasis, yet this is not a prominent feature of this manuscript.

The maunscript deals with most important factors involved in angiogenesis. But the reviewer was confused by findig the most prominent factors and issues at the end of the manuscript. Therefore rearranging the sections or at least including an layout/explanation at the end of the introduction would greatly improve the manuscript.

The literature focuses on recent publications, but ommits major classical papers, which would exactly fit statements made the introduction. In addition, the long history of tumorangiogenesis reseach including treatment approaches and especially the not so successfull translation into the clinic should be taken into account.

Figure(s) giving an graphic overview on the functions of the various factors and table(s) on their use as biomarkers (with tumor types) and/or targets of specific therapies (includig the substances used as therapeutics and clinical results). Both roles should be clearly distinguished.

Yet the only Figures are on the authors own previous publications on CD34 in lymphnodes and a very simplistic graphic model on a specific part of tumorangiogenesis.

The last section on non-reliability recalculated data given in two publications. Apparently both on histological tumor evaluations. While poiniting out false data interpretation is important, the conclusion that ‚unreliable results appear at all stages of the study of tumor vascularization‘ is too general. The abstracts reads more like an introduction than a summary of the manuscript. This must be improved.

The conclusions should contain more specifics.

The terms ‚angiogenesis‘ and vascularisation‘ should be presicely defined and used.

It should be clearly stated at the beginning of section 2, that angiogenesis in lymphnodes is not lymphangiogenesis. This would help to focus readers attention on CD34 as an potential biomarker, and make it clear, that this manuscript is not about tumor cell spread via lymphatic vessels. This reviewer suggests to shorten this section.

Figures 1 and 2 should be combined into one figure, if still included in the final manuscript. Layout needs reworking. The figures could be improved, by indicating the specific areas discribed in the legend. This could better explain the differences in staining intensity and tissue structure to readers. The change in vessel morphology and size (lenght) should be commented on. Scale bars must be correctly allingned.

The in depth discussion of female reproductive cancers (a whole section) should be highlighted early on (abstract and introduction)  

English expressions should be checked. It would majorly improve the manuscript, if sentences would usually concentrate on one argument/statement.

Usually in the authors list, the exact job description is not necessary.

Author Response

I am grateful to the Dear Reviewer for the work and time spent in studying our manuscript. Regarding the comments and recommendations made:0

The abstract makes this revier expect a focus on the spacial aspects of angiogenesis within a tumor and between tumor and metastasis, yet this is not a prominent feature of this manuscript.

The abstract has been refined

The maunscript deals with most important factors involved in angiogenesis. But the reviewer was confused by findig the most prominent factors and issues at the end of the manuscript. Therefore rearranging the sections or at least including an layout/explanation at the end of the introduction would greatly improve the manuscript.

We believe that the most important thing in this branch of oncology is the study of the prognostic value of angiogenesis, as well as the factors influencing this process, activating or suppressing it. Therefore, these sections are placed at the very beginning of the manuscript.

In connection with the recommendations of the Dear Reviewer, a short list of sections of the manuscript has been included in the "Introduction".

The literature focuses on recent publications, but ommits major classical papers, which would exactly fit statements made the introduction. In addition, the long history of tumorangiogenesis reseach including treatment approaches and especially the not so successfull translation into the clinic should be taken into account.

Due to the fact that we tried to study the most recent publications as fully as possible, the long history of the study of angiogenesis, unfortunately, remained outside the manuscript. The article briefly discusses some pharmacological preparations that suppress angiogenesis and the significant side effects of these drugs. A detailed description of these drugs, their positive effects and complications, may be the purpose of a separate literature review.

In addition, the manuscript is formatted as a "Literature Review", so we have tried to include articles with the results of original research as fully as possible. Whereas the main classic articles describing the state of the vessels in the tumor are mainly also reviews of the literature.

Figure(s) giving an graphic overview on the functions of the various factors and table(s) on their use as biomarkers (with tumor types) and/or targets of specific therapies (includig the substances used as therapeutics and clinical results). Both roles should be clearly distinguished.

Yet the only Figures are on the authors own previous publications on CD34 in lymphnodes and a very simplistic graphic model on a specific part of tumorangiogenesis.

Unfortunately, I draw very badly, I can't draw at all. This is my feature. Therefore, I was able to create a simple graphic model and supplemented the manuscript with my "old" figures, which shows what looks like the formation at cancer of BLOOD vessels in the lymph nodes  without metastases.

The last section on non-reliability recalculated data given in two publications. Apparently both on histological tumor evaluations. While poiniting out false data interpretation is important, the conclusion that ‚unreliable results appear at all stages of the study of tumor vascularization‘ is too general.

Tried to fix

The abstracts reads more like an introduction than a summary of the manuscript. This must be improved.

Tried to fix

The conclusions should contain more specifics.

Tried to fix

The terms ‚angiogenesis‘ and vascularisation‘ should be presicely defined and used.

I know well what is the difference between angiogenesis, vasculogenesis and vascularization. However, in the articles studied, authors often report angiogenesis or vasculogenesis when describing an increase in the number of vessels.

Tried to correct all text.

It should be clearly stated at the beginning of section 2, that angiogenesis in lymphnodes is not lymphangiogenesis. This would help to focus readers attention on CD34 as an potential biomarker, and make it clear, that this manuscript is not about tumor cell spread via lymphatic vessels. This reviewer suggests to shorten this section.

Lymphangiogenesis in the lymph nodes is not described, since in ontogenesis or in the time of recovery after injury, the lymph nodes originate from the lymphatic vessels, so all mention of angiogenesis in these organs is associated with an increase in blood vascularization, intranodal blood vessels. It seems to me that it is very interesting to test the idea that emerged from the remark of the Dear Reviewer about the appearance of lymphatic vessels in the lymph nodes at cancer in the region of the lymph collection.

It seems to us that the optimal marker for detecting newly formed blood vessels is the CD34 antigen. Antibodies to this antigen make it possible to detect not all vessels, namely, newly formed, "young" ones.

This paragraph has been changed.

Figures 1 and 2 should be combined into one figure, if still included in the final manuscript. Layout needs reworking. The figures could be improved, by indicating the specific areas discribed in the legend. This could better explain the differences in staining intensity and tissue structure to readers. The change in vessel morphology and size (lenght) should be commented on. Scale bars must be correctly allingned.

Each figure depicts one of the areas indicated in the legend. In this case, we do not aim to show and discuss the change in the vessel morphology. We are only trying to show how these newly formed, young vessels look.

Tried to fix

The in depth discussion of female reproductive cancers (a whole section) should be highlighted early on (abstract and introduction)  

Tried to fix

English expressions should be checked. It would majorly improve the manuscript, if sentences would usually concentrate on one argument/statement.

I can only apologize for the quality of the English language. But the translation was made by a certified translation agency, and then revised by a US citizen.

Usually in the authors list, the exact job description is not necessary.

Left unchanged

Once again, I thank the Dear Reviewer for studying the manuscript, and I hope that we have answered all of his questions and comments.

Round 2

Reviewer 2 Report

This reviewer thanks the authors for the reply on the reviewers comments, even if made in a very explicite way.

Some issues have been adressed, several still remain unchanged. 

Especially (now) Figure 2 still is not of the same scientific standard as most of the manuscript. 

Enhancement of the english language still would improve the manuscript (this includes the title).

In acknowledging the authors replies, this reviewer explicitly leaves it to the editors to decide what changes requested in the first round are adequatly adressed and where additional work is required in order to publish this manuscript.

Author Response

I once again thank the Dear Reviewer for the time devoted to studying our manuscript. I also once again apologize if I was harsh and overconfident during the peer review process and scientific discussion.

Figure 2 remains in the manuscript in agreement with the Academic Editor.

The title of the article has been changed.

I sent the manuscript to the University (to the Department of English) to check the quality of the translation.